# IGF1R/IR Mediates Resistance to BRAF and MEK Inhibitors in BRAF-Mutant Melanoma

**DOI:** 10.3390/cancers13225863

**Published:** 2021-11-22

**Authors:** Hima Patel, Rosalin Mishra, Nour Yacoub, Samar Alanazi, Mary Kate Kilroy, Joan T. Garrett

**Affiliations:** 1Department of Pharmaceutical Sciences, College of Pharmacy, University of Cincinnati, Cincinnati, OH 45267, USA; patel2h2@mail.uc.edu (H.P.); mishrarn@ucmail.uc.edu (R.M.); alanazsa@mail.uc.edu (S.A.); kilroymk@mail.uc.edu (M.K.K.); 2College of Medicine, Northeast Ohio Medical University, Rootstown, OH 44264, USA; nyacoub@neomed.edu

**Keywords:** melanoma, BRAF, resistance, IGF1R, IR

## Abstract

**Simple Summary:**

Melanoma accounts for only 4% of skin cancer, but is the major cause of skin cancer related deaths. The use of dabrafenib (BRAF inhibitor) and trametinib (MEK inhibitor), two FDA approved drugs to treat patients with BRAFV600E melanoma, is limited in the clinic due to the development of resistance. The IGF family of receptors is known to play a crucial role in cancer progression. In our in vitro screening, we identified that the activation of Insulin-like growth factor 1 receptor (IGF1R) and Insulin Receptor (IR) mediates resistance to dabrafenib and trametinib. Patients with high levels of IGF1R and IR have worse survival outcomes compared to patients with low levels of these receptors. We demonstrate that combining dabrafenib and trametinib with an IGF1R/IR inhibitor, BMS-754807, in vitro and in vivo, is efficacious and inhibits proliferation and tumor growth. This research opens up avenues for the development of novel and potent IGF1R/IR inhibitors for patients with BRAF-mutant melanoma.

**Abstract:**

The use of BRAF and MEK inhibitors for patients with BRAF-mutant melanoma is limited as patients relapse on treatment as quickly as 6 months due to acquired resistance. We generated trametinib and dabrafenib resistant melanoma (TDR) cell lines to the MEK and BRAF inhibitors, respectively. TDR cells exhibited increased viability and maintenance of downstream p-ERK and p-Akt as compared to parental cells. Receptor tyrosine kinase arrays revealed an increase in p-IGF1R and p-IR in the drug resistant cells versus drug sensitive cells. RNA-sequencing analysis identified IGF1R and INSR upregulated in resistant cell lines compared to parental cells. Analysis of TCGA PanCancer Atlas (skin cutaneous melanoma) showed that patients with a BRAF mutation and high levels of IGF1R and INSR had a worse overall survival. BMS-754807, an IGF1R/IR inhibitor, suppressed cell proliferation along with inhibition of intracellular p-Akt in TDR cells. Dual inhibition of IGF1R and INSR using siRNA reduced cell proliferation. The combination of dabrafenib, trametinib, and BMS-754807 treatment reduced in vivo xenograft tumor growth. Examining the role of IGF1R and IR in mediating resistance to BRAF and MEK inhibitors will expand possible treatment options to aid in long-term success for BRAF-mutant melanoma patients.

## 1. Introduction

Melanoma, the most aggressive form of skin cancer, originates in melanin producing melanocytes of the skin [1]. The five-year survival rate for patients diagnosed with metastatic melanoma is around 20% [2]. The mitogen-activated protein kinase pathway (MAPK), comprised of downstream effector proteins Ras, MEK, and ERK, plays a pivotal role in the proliferation and mutagenesis in melanoma [3]. The PI3K/Akt pathway and the loss of tumor suppressor PTEN contribute to melanoma tumorigenesis [4,5,6]. Up to 90% of melanomas contain BRAF^T1799A^ transversion, which encodes for constitutively active BRAF^V600E^ oncoprotein. This is critical for the survival and proliferation of melanoma cells and results in hyperactivation of the MAPK pathway [7]. Co-targeting BRAF and MEK with dabrafenib and trametinib, respectively, demonstrated a long term benefit and dramatically improved response rates in about 70% patients harbouring the BRAF mutation as compared to standard chemotherapy and single agent BRAF inhibitors [8,9].

Chronic treatment with these inhibitors has met challenges as patient responses begin to drop due to the development of acquired resistance. Resistance to BRAF inhibitor vemurafenib or dabrafenib monotherapy occurs in 50% of patients within 6–8 months of drug treatment [10,11]. Disease progression for BRAF-mutant metastatic melanoma patients treated with a combination of dabrafenib and trametinib occurs within 9–10 months due to the development of resistance [12,13,14]. Resistance to BRAF and MEK inhibitors include, but are not limited to, alterations in BRAF splicing [15], mutations in NRAS [16], BRAF copy number amplification [17], activation of PI3K/Akt pathway, and hyperactivation of receptor tyrosine kinases (RTK), cell surface receptors that activate the MAPK and PI3K/Akt pathways [18,19,20].

IGF1R and IR are RTKs that belong to the insulin-like growth factor (IGF) family of receptors. These receptors are commonly overexpressed and their role in tumor cell proliferation and survival is well established in various malignancies [21,22,23,24,25], including melanoma [26]. Overexpression leads to alterations in the levels of regulators of these receptors, including ligands IGF1, IGF2, insulin, and serum insulin-like growth factor binding proteins (IGFBP 1–6). Activation of these receptors mediates downstream MAPK, PI3K/Akt, or pathways associated with cell contraction and motility by recruiting various adaptor proteins such as insulin receptor substrate 1 (IRS1), IRS2, and Src homology domain containing (Shc) [27,28]. Dysregulation of the IGF1R pathway acts as an oncogenic signal in initial tumorigenesis as well as resistance to targeted therapies [29].

In this study, we generated dabrafenib and trametinib resistant melanoma cell lines and characterized the cells. We observed that ERK and Akt were constitutively active in resistant cells as compared to the parental counterparts. We demonstrate that IGF1R and IR are activated in response to chronic dabrafenib and trametinib resistance in BRAF-mutant melanoma and are differentially expressed in parental and resistant cell lines. We utilized BMS-754807, a potent IGF1R/IR inhibitor with limited off targets effects [30,31]. We observed that the MAPKi resistant melanoma cells treated with BMS-754807 decreased the short and long-term proliferation along with the inhibition of phosphorylation of Akt. In vivo in a subcutaneous xenograft model, dabrafenib and tramatenib resistant cells demonstrated increased tumorigenicity. Administration of BMS-754807 along with dabrafenib and trametinib significantly suppressed tumor growth compared to vehicle or dabrafenib and trametinib treatment. These findings provide mechanistic insight and a rationale for targeting MAPKi resistance BRAF-mutant melanoma with IGF1R/IR inhibition.

## 2. Materials and Methods

### 2.1. Reagents

Dabrafenib and trametinib were purchased from LC Laboratories. BMS-754807, Linsitinib, BMS-536924, and GSK1838705A were purchased from MedChemExpress. SCH772984 was obtained from Selleckchem (Houston, TX, USA). All drugs were dissolved in DMSO (Fisher Scientific(Waltham, MA, USA)) at described concentrations and stored at −20 ℃. IGF2 (cat#100-12) was purchased from Peprotech (Cranbury, NJ, USA). 

### 2.2. Cell Lines and Culture Conditions

A375 and WM115 cells were purchased from American Type Cell Culture (ATCC, Manassas, VA, USA). WM983B cells were purchased from Rockland Immunochemicals, Inc (Limerick, PA, USA). A375 cells were cultured in DMEM (Corning, cat#MT15017CV); WM115 cells were cultured in MEM (Corning, cat#MT10009CV); WM983B cells were cultured in RPMI-1640 medium (Corning, cat#MT10040CM) supplemented with 10% (vol/vol) FBS (VWR, cat# 89510-188) and 1% Penicillin–streptomycin solution 100× (Corning, cat#30-002-CI). Cultured cells were incubated in a humidified incubator at 37 °C with 5% (vol/vol) CO_2_ and 95% (vol/vol) air.

### 2.3. Establishing Resistant Lines

A375, WM983B, and WM115 trametinib and dabrafenib resistant lines (TDR cells) were generated by treating parental cells with increasing concentration of dabrafenib and trametinib starting over a period of 9–11 months until cells could grow in the presence of a specified concentration of dabrafenib and trametinib. A375 TDR cell lines were maintained in 250 nM dabrafenib and 12.5 nM trametinib. WM983B TDR cells were maintained in 2.4 μM dabrafenib and 500 nM trametinib. WM115 TDR cells were maintained in 800 nM dabrafenib and 200 nM trametinib. The TDR cells and corresponding parental cells were tested for mycoplasma contamination and authenticated by STR profiling by ATCC every six months.

### 2.4. RTK Array

Parental and TDR cells treated with DMSO or a combination of dabrafenib and trametinib were lysed and 400 µg proteins were used for Proteome Profiler Human Phospho-RTK Array Kit (R&D Systems, cat#ARY001B(Minneapolis, MN, USA)) as per the manufacturer’s instructions. The intensities on the array were quantified using a negative control on the array and Image J software 1.8.0 (NIH) (Bethesda, MD, USA).

### 2.5. Cell Proliferation Assay and Long Term Growth Assays

#### 2.5.1. MTT Assay

Growth kinetics of A375, A375 TDR, WM983B WM983B TDR, WM115, and WM115 TDR cells were determined by the 4,5-dimethylthiazol-2-yl)-2,5-diphenyltetrazolium bromide (MTT) assay. Briefly, 1 × 10^4^ cells/well were seeded in 96-well plates in triplicates. For Figure 1B, a total of 10 concentrations of dabrafenib and trametinib were used in combination. The concentrations of dabrafenib and trametinib for WM115 were vehicle, 0.1 nM, 0.5 nM,1 nM, 5 nM, 10 nM, 50 nM, 100 nM, 500 nM, and 1 μM. The concentrations of dabrafenib and trametinib for WM983B were vehicle, 0.5 nM, 1 nM, 5 nM, 10 nM, 50 nM, 100 nM, 500 nM, 1 μM, and 5 μM. The concentrations of dabrafenib and trametinib for A375 were vhicle, 0.1 nM, 0.5 nM, 1 nM, 5 nM, 10 nM, 50 nM, 100 nM, 500 nM, and 1 μM. For Figure 1C, a total of 10 concentrations of dabrafenib and trametinib used were as follows: vehicle, 0.1 nM, 0.5 nM, 1 nM, 5 nM, 10 nM, 50 nM, 100 nM, 500 nM, and 1 μM. After 72 h of treatment with dabrafenib and trametinib, the medium was substituted with 0.5 mg/mL MTT solution (Sigma Aldrich) for 4 h and absorbance was recorded at 570 nm using SPECTRAmax PLUS Microplate Spectrophotometer Plate Reader (Molecular Devices Corporation, San Jose, CA, USA) and expressed as the mean of triplicates relative to vehicle (DMSO) control together with standard error of mean (SEM). IC50 values were generated using the GraphPad Prism 7 software, version 7.01 (San Diego, CA, USA). In separate experiments, dabrafenib and trametinib were withdrawn from the TDR cells for 21 days and an MTT assay was performed with parental, TDR, and TDR cells with drugs withdrawn from them in the presence of dabrafenib and trametinib with the same conditions as mentioned above. Growth kinetics of TDR cells in the presence of 3, 5, and 10 μM BMS-754807 were assessed as mentioned above.

#### 2.5.2. Crystal Violet Assay

TDR cells were seeded at a density of 2 × 10^4^ cells/well in 12 well plates (Falcon) in triplicates. Complete media containing BMS-754807 (5 μM), dabrafenib, and trametinib was replaced every 2 days. On day 21, cells were stained with 0.5% crystal violet. The cultures were exposed to crystal violet solution for 2 min and washed off with water to remove the excess stain. The intensities were measured using an Odyssey infrared system (LI-COR, Lincoln, NE, USA). The Experiments were repeated at least twice.

#### 2.5.3. Matrigel Growth Assay

Three-dimensional (3D) growth assays were performed in growth factor-reduced matrigel (BD Biosciences, cat#CB-40230 (San Jose, CA, USA)) where 96-well plates were coated with 70 µL of matrigel/well. TDR cells (2 × 10^4^/well) were plated and incubated at 37 °C for 24 h. Cells were treated with vehicle (DMSO), BMS-754807 (5 μM) and/or dabrafenib + trametinib every alternate day. After 21 days of incubation, cells were visualized and photographs were captured from 5 random fields under microscope (Nikon, Melville, NY, USA) at 10× magnification. The areas of the cells were measured by ImageJ and represented as mean areas normalized to DMSO control.

### 2.6. Western Blot Analysis

Cells were lysed in RIPA buffer (Thermo Fisher Scientific, Cat#BP-115) supplemented with protease and phosphatase inhibitors (Thermo Fisher Scientific, Cat#88669). Protein concentration was determined using the Pierce BCA Protein Assay Kit (Thermo Scientific, Cat#23225) according to manufacturer’s instructions. A minimum of 15 µg of protein was used for each sample. Primary antibodies used for immunoblotting are described in Appendix A. The primary antibody (1:1000 dilution) was added to the nitrocellulose and incubated overnight at 4 °C with gentle agitation. Blots were then washed with 1x TBST (3 washes, 10 min each) and incubated with anti-rabbit IgG-horse radish peroxidase (HRP) (Santa Cruz Biotechnology, cat#SC-2357; 1:2000 dilution) for 1 h at room temperature. Blots were then washed with 1X TBST (3 washes, 10 min each). Protein signals were detected using Pierce ECL Western blotting substrate (Thermo Fisher Scientific, Cat# 32106). For serum starvation experiment, TDR cells were serum starved for 24 h and IGF2 (10 µg/mL) was added at specified time points. In a separate set of experiments, cells were serum starved for 24 h, followed by addition of BMS-754807 and IGF2. No blots were stripped and reprobed.

For lysing mice tumors, the tumors were thawed on ice and RIPA buffer supplemented with 3 times concentration of protease and phosphate inhibitors was added for 20 min. The tumor was homogenized using a homogenizer for 5 min, briefly spun in centrifuge, and the supernatant was collected. Protein concentration was determined and Western blot analysis was performed. No blots were stripped and reprobed.

### 2.7. RNA Extracztion and RT-qPCR

Total RNA was extracted from parental and TDR cells treated with DMSO or dabrafenib + trametinib using the RNA extraction mini kit (Qiagen, cat#74104, Hilden, Germany) according to the manufacturer’s instructions. The RNA concentration was determined using a NanoDrop™ (ThermoScientific, Waltham, MA, USA) spectrophotometer. Two-step RT-qPCR was performed to assess the mRNA levels. First strand cDNA was synthesized using iScriptTM cDNA Synthesis Kit (BioRad, cat#1708891, Hercules, CA, USA). The mRNA levels were quantified by qPCR using primers purchased from Integrated DNA Technologies. qPCR was set up using CFX96 Real-Time System (BioRad). The relative mRNA levels were calculated by the 2^−ΔΔCt^ method. Actin was used as internal control. Primers used for the experiment are listed in Appendix A.

### 2.8. Knockdown of IGF1R and INSR

TDR cells were transfected with 20 nM of siRNA, specifically targeting ON-TARGETplus Human IGF1R siRNA-SMARTPool IGF1R (Horizon Discovery) (Sequences: GAACAAGGCUCCCGAGAGU; AAACGAGGCCCGAAGAUUU; ACGGAGACCUGAAGAGUGCUA; GCAGGUCCCUUGGCGAUGU) and/or ON-TARGETplus Human INSR siRNA-SMARTpool (Horizon Discovery) and ON-TARGETplus Non-targeting Pool (Horizon Discovery, Cambridge, England) (Sequences: GGAAGCACCCUUUAAGAAU; GGACUCAGUACGCCGUUUA; AAAUACGGAUCACAAGUUG; AGUGAGAUCUUGUACAUUUC) as control as instructed by manufacturer’s protocol. After 72 h, cells were harvested for proliferation and Western Blot analysis.

### 2.9. RNA Sequencing

Total RNA was extracted using the RNA extraction mini kit (Qiagen, cat#74104) according to the manufacturer’s instruction. The RNA quality was determined by Bioanalyzer (Agilent, Santa Clara, CA, USA). To isolate the polyA RNA, NEBNext Poly (A) mRNA Magnetic Isolation Module (New England BioLabs, Ipswich, MA, USA, cat#E7490S) was used with a total of 1 µg of good quality total RNA as input. The NEBNext Ultra II Directional RNA Library Prep Kit (New England BioLabs, cat#E7760S) was used for library preparation, which is dUTP-based stranded library. After library quality control and quantification, individually indexed and compatible libraries were proportionally pooled and sequenced using the Illumina HiSeq 1000 sequencing platform. Under the sequencing setting of single read 1 × 51 bp, about 25 million pass filter reads per sample were generated. Data were analyzed using iGEAK [32] (Interactive gene expression analysis kit; Cincinnati Children’s Hospital and Medical Center, Cincinnati, OH, USA).

### 2.10. In Vivo Xenograft Study

Institutional Animal Care and Use Committee (IACUC), University of Cincinnati ethically approved the in vivo mouse experiment (UC IACUC approval: 19-04-19-01; Approval date: 14 April 2021). Male athymic nude mice were purchased from Jackson Laboratories at the age of 3–4 weeks (cat# 007850 Homozygous Foxn1nu). Approximately 1 × 10^6^ cells WM115 TDR cells resuspended in 1× PBS and growth factor-reduced matrigel (BD Biosciences, cat#CB-40230) (1:1 ratio, total volume injected 200 µL) were injected s.c. into right flank of the mice. Formice injected with WM115 parental and TDR cells, the cells were prepared as mentioned above and WM115 parental were injected into the left flank and WM115 TDR cells were injected into the right flank of nude mice. Tumor volume was determined using the formula: tumor volume = length × width^2^/2. For WM115 TDR tumors, when the tumors reached 200 mm^3^, mice were randomized into three groups: (i) vehicle (PEG400:water 1:1 and 0.5% HPMC and 0.2% Tween; (ii) dabrafenib (30 mg/kg daily) + trametinib (0.3 mg/kg daily) resuspended in 0.5% HPMC and 0.2% Tween and (iii) BMS-754807 (10 mg/kg twice daily; resuspended in PEG400: water 1:1 ratio) and dabrafenib (30 mg/kg daily) and trametinib (0.3 mg/kg daily) resuspended in 0.5%HPMC and 0.2% Tween. The drugs were administered by p.o. route using an oral gavage needle. The volume of drug administered was 100 µL. The duration of treatment was 15 days. Vehicle and dabrafenib + trametinib were administered once a day in the morning. Two doses of BMS-754807 administered in a day were at least 8 h apart, once in the morning and once in the evening. Tumor size was recorded 2 to 3 times a week post randomization. The last dose of drugs was given 1 h prior to sacrificing mice. Tumors were collected and snap-frozen in liquid nitrogen and placed in −80 ℃ and then used for Western blot analysis.

### 2.11. Statistical Analysis

Data are shown as the mean ± standard error of mean (SEM) and representative of at least three independent experiments unless otherwise indicated. Statistical analysis among groups was performed using the two-tailed Student’s *t*-test, and one-way analysis of variance; *p* < 0.05 was considered statistically significant. Logrank Test *p*-value for patient data was generated by cBioPortal.

## 3. Results

### 3.1. Phospho-ERK Levels Are Maintained in Response to Chronic Dabrafenib and Trametinib Resistance In Vitro

We exposed BRAF mutant melanoma cell lines A375, WM115, and WM983B to increasing concentrations of the BRAF inhibitor dabrafenib and the MEK inhibitor trametinib (Figure 1A). The BRAF status of these cell lines along with PTEN and CDKN2A status are summarized in Table 1.

We generated trametinib- and dabrafenib-resistant (TDR) WM115, A375, and WM983B cell lines via a gradual dose escalation of each drug and found a shift in IC_50_ values for resistant cells (Figure 1B). When parental cells were treated with dabrafenib, trametinib, or a combination of both inhibitors, we observed a striking reduction in acini size as compared to TDR cells (Appendix A). We observed that with three weeks of drug withdrawal, drug resistant cells maintained their drug resistant phenotype, (Figure 1C). The parental drug sensitive cell lines demonstrated reduced p-Erk in the presence of dabrafenib and/or trametinib, whereas drug resistance cells maintained p-Erk in the presence of drug (Figure 1D). Furthermore, drug resistant cells maintained p-Erk in the presence of the Erk1/2 inhibitor SCH772984 (Figure 1E). These data indicate that the drug resistant cells maintain p-Erk in the presence of drug and preserve their drug resistance phenotype after three weeks of drug withdrawal.

### 3.2. IGF1R and IR Are Activated and Upregulated in BRAF-Mutant Melanoma TDR Cell Lines

We next examined if there was differential phosphorylation of receptor tyrosine kinases (RTK) between the parental and drug resistant cells by performing RTK arrays which examine the phosphorylation status of 49 different RTKs. We observed that drug resistant cells treated with drugs had elevated levels of phosphorylated EGFR, HER3 (ErbB3), IR, IGF1R, Axl, and Dtk (Figure 2A). We quantified the signal density and observed that phosphorylated HER3 was increased in drug resistant WM115 cells treated with vehicle DMSO or dabrafenib and trametinib treatment (Figure 2B). Phosphorylated Axl was increased in drug resistant WM115 cells treated with vehicle DMSO and in drug resistant WM983B cells treated with vehicle DMSO or dabrafenib and trametinib treatment. Phosphorylated EGFR was increased in drug resistant WM983B cells treated with vehicle DMSO or combination of dabrafenib and trametinib. Phosphorylated Dtk was increased in drug resistant WM983B cells treated with vehicle DMSO. Phosphorylated IGF1R was increased in drug resistant WM115 and A375 cells treated with vehicle DMSO or dabrafenib and trametinib treatment. Strikingly, phosphorylated IR was increased in all three drug resistant cells treated with vehicle DMSO or dabrafenib and trametinib treatment. Next, to validate the RTK array results, we performed a Western blot analysis (Figure 2C). Phosphorylated HER3 was increased in drug resistant WM115 cells treated with vehicle DMSO or dabrafenib and trametinib treatment compared to parental DMSO treated cells. Phosphorylated Axl was increased in drug resistant WM115 and WM983B cells, correlating with RTK array data. Phosphorylated EGFR was increased in drug resistant WM983B cells treated with vehicle DMSO or dabrafenib and trametinib. Addition of dabrafenib and trametinib to WM115, WM983B, and A375 parental cells had no effect on p-IGF1R/p-IR levels compared to parental cells treated with DMSO. In accordance with the results from RTK array, WM115 and A375 TDR cells treated with DMSO or dabrafenib and trametinib exhibited an increase on phosphorylation at p-IGF1R at site Y1131 and p-IR at site Y1146, while these levels were maintained in WM983B TDR cells. These data indicate that drug resistant cells display differential phosphorylation of RTKs with notably increased phosphorylated IGF1R/IR.

### 3.3. Differential Expression of IGF1R and INSR in Parental and TDR Cells

We performed RNA-seq to examine differences between BRAF pathway inhibitor drug sensitive and drug resistant cells at the gene level. We found that IGF1R is upregulated in all three drug resistant cells compared to parental cells (Figure 3A). INSR, the gene that encodes IR, was increased in drug resistant A375 and WM115, but not in WM983B cells. IGF1 and IGF2, the high affinity ligands to IGF1R, were increased in drug resistant WM115 cells only. Addition of dabrafenib and trametinib to parental cells led to the upregulation of INSR in all three parental cell lines. IGF1R was only upregulated in A375 parental cells treated with dabrafenib and trametinib and not in WM983B and WM115 cell lines. Treatment with dabrafenib and trametinib for WM983B and WM115 parental cells increased the levels of IGF1 and IGF2. A375 parental cells treated with dabrafenib and trametinib exhibited downregulation of their ligands, IGF1 and IGF2. The differences in gene expression across the three cell lines could be attributed to their characteristics, i.e., WM983B is a metastatic cell line whereas A375 and WM115 are primary melanoma cell lines along with the presence of various mutations described in Table 1. As a separate way to quantify RNA levels, we performed real-time quantitative PCR for IGF1R, IGF1, IGF2, and INSR (Figure 3B). Consistent with RNA-seq data, we found that IGF1R increased in all TDR compared to parental cells. INSR was increased in drug resistant A375 and WM115, but not in WM983B cells, correlating with RNA-seq data. We found that IGF1 and IGF2 levels increased in drug resistant WM115 cells only, but not in WM983B and or A375 cells. Treating WM115 and WM983B parental cells with dabrafenib and trametinib significantly decreased the mRNA expression of IGF1R and INSR, while the expression of IGF1R was increased for A375 parental cells under the same condition. IGF1 expression was increased when WM983B parental cells were treated with a combination of dabrafenib and trametinib. Addition of dabrafenib and trametinib to A375 parental cells augmented IGF2 levels by two folds. We observed an increase in mRNA expression for Axl, Protein S, and Gas6 in TDR cell lines (Appendix A). We next examined that survival curves of melanoma patients from the TCGA data set for patients with a BRAF mutation along with high or low levels of INSR or IGF1R. Patients with high levels of INSR had a statistically significant worse survival compared to patients with low levels of INSR (Figure 3C). Comparison of survival for patients with high levels of IGF1R versus low levels of IGF1R approached significance with a *p*-value of 0.0723. Overall, these data indicate that the BRAF pathway inhibitor resistant cells have elevated IGF1R at the mRNA level and melanoma patients with elevated IGF1R and INSR have a worse survival compared to patients with low levels of IGF1R.

### 3.4. BMS-754807 Inhibits Cell Proliferation and Intracellular Akt Signaling in TDR Cells

We probed the effect of pharmacological inhibition of IGF1R and IR in our BRAF pathway inhibitor resistant cells using BMS-754807, a reversible inhibitor of IGF1R and IR. While the addition of BMS754807 to dabrafenib and trametinib resulted in a reduction in proliferation (Figure 4A), Linsitinib, BMS-536924, and GSK1838705A failed to inhibit proliferation in combination with dabrafenib and trametinib in TDR cells (Appendix A). BMS-754807 as a single agent did not inhibit cell proliferation and p-IGF1R/p-IR expression in TDR cells (Appendix A). A 21-day crystal violet assay revealed similar results (Figure 4B). We next plated BRAF pathway inhibitor resistant cells on a basement membrane of matrigel and treated with vehicle, dabrafenib and trametinib or dabrafenib and trametinib and BMS-754807. We observed a striking reduction in size of acini in cells treated with dabrafenib and trametinib and BMS-754807 (Figure 4C). We assessed the effect that BMS-754807 has on signaling in drug resistant cells. After stimulation with the ligand IGF2, BMS-754807 treatment reduced p-IGF1R, p-IR, and p-Akt, but did not affect p-Erk levels (Figure 4D; Appendix A). These data indicate that pharmacological inhibition of IGF1R and IR reduces the growth of BRAF pathway inhibitor resistant cells and reduces p-IGF1R, p-IR, and p-Akt within these cells.

### 3.5. Knockdown of IGF1R/INSR Regulates TDR Cell Viability

We next assessed the effect of genetic inhibition of IGF1R and IR in our BRAF pathway inhibitor resistant cells. We found that siRNA targeting IGF1R reduced IGF1R in all three cell lines (Figure 5A). INSR siRNA reduced IR levels and the combination of IGF1R and INSR siRNA reduced both IGF1R and IR in all three cell lines. We performed an MTT cell proliferation assay on cells after transfection with siRNA targeting IGF1R, INSR, or both siRNAs. We observed a reduction in the proliferation for cells transfected with siRNA targeting IGF1R and INSR compared to siRNA control. siRNA targeting INSR reduced cell proliferation of A375 TDR cells (Figure 5B). These data indicate genetic inhibition of IGF1R and IR reduces the growth of BRAF pathway inhibitor resistant cells. 

### 3.6. IGF1R/IR Blockade Decreases Tumor Growth in WM115 TDR Xenograft

We assessed the in vivo growth potential of WM115 parental and BRAF pathway inhibitor resistant cells. Athymic nude mice were injected on the right flank with WM115 TDR and the left flank with parental WM115 cells (Figure 6A). We observed that drug resistant cells grew over time whereas the parental drug sensitive cells did not grow in our hands (Figure 6B). These data indicate the growth potential in vivo of the drug resistant cells. We next performed a drug efficacy study in WM115 TDR tumors. Once tumors were greater than 200 mm^3^, mice were randomly allocated to either vehicle, dabrafenib and trametinib, or dabrafenib and trametinib and BMS-754807. At day 15 of treatment, mice treated with dabrafenib and trametinib and BMS-754807 had a statistically significant reduction in tumor growth compared to either vehicle or dabrafenib and trametinib treated groups (Figure 6C). We did not observe a significant change in weight of mice treated with dabrafenib and trametinib and BMS 754807 (Figure 6D). Mice were given drug one hour before sacrifice at day 16, sacrificed, and tumors were harvested for analysis. Western blot analysis of xenografts revealed a reduction in p-IGF1R/p-IR in tumors treated with dabrafenib and trametinib and BMS-754807 compared to vehicle or dabrafenib and trametinib treated tumors (Figure 6E). Downstream of p-IGF1R/p-IR, we observed a reduction in p-Akt in two out of three tumors treated with dabrafenib and trametinib and BMS-754807 compared to vehicle or dabrafenib and trametinib treated tumors. These data indicate that inhibition of IGF1R/IR could be an effective strategy to overcome resistance to BRAF pathway inhibitor resistant tumor. All original images from Western blots are included in Appendix A.

## 4. Discussion

Acquired resistance to dabrafenib and trametinib occurs in most patients with BRAF-mutant metastatic melanoma and the majority of these patients fail the targeted therapy regimen. About 20% of these acquired resistant melanomas harbor alterations that stimulate and activate downstream MAPK and PI3K/Akt signaling pathways [36]. While BRAFV600E amplification along with non-MAPK alteration tend to co-occur along with other genetic changes within resistant tumors, mutations in MEK1/2, BRAF, and NRAS occur in isolation [7,12,37]. In order to mimic the resistance setting in vitro, we generated trametinib and dabrafenib resistant cell (Figure 1A). The final concentration of dabrafenib and trametinib used to generate the TDR cells was dictated by the sensitivity of the parental cell lines to the combination, which could be attributed to the type of BRAF mutation present in the cell line along with other mutations such as CDKN2A and PTEN status. It is well established that MAPK reactivation occurs in resistance to the combination of dabrafenib and trametinib in BRAF-mutant melanoma [38,39]. Likewise, we observed that p-Erk levels were maintained in TDR cells in response to dabrafenib and/or trametinib (Figure 1D). SCH772984, an ERK1/2 inhibitor, failed to show any effect on TDR cells (Figure 1E), which incited us to explore other mechanisms of resistance.

IGF1R and IR are frequently expressed at high levels in various malignancies such as breast cancer [40,41,42], colorectal cancer [43,44], and NSCLC [45,46]. Dysregulation of IGF1R/IR axis acts as an oncogenic signal in initial tumorigenesis as well as mediating resistance to targeted therapies [47,48]. The IGF2/IGF1R/IR axis mediates adaptive resistance to erlotinib by undergoing an IGF1R/IR phenotypic switch in cholangiocarcinoma [49]. A recent study identified that a single nucleotide polymorphism (SNP) of IGF1R may have a protective effect for melanoma risk [50] in support that differential IGF1R signaling is involved in melanoma development. A phosphoproteomic screen identified IGF1R and IR as a new therapeutic opportunity for driver inhibition while simultaneously preventing the development of acquired resistance [51,52]. Paired tissue samples from BRAF inhibitor relapsed patients exhibited increased IGF1R expression [53]. A recent study has also shown that upregulation of IGF1R in BRAF and MEK inhibitor resistant cells involves the activation of the MEK5-ERK5 pathway [54]. IGF1R, along with IR, was activated and upregulated in TDR cells as compared to the parental cells and this was confirmed by RTK array analysis (Figure 2A–C). Upregulation (WM115 and A375TDR) or maintenance (WM983B TDR) of p-IGF1R and p-IR levels could be a function of basal levels of these proteins present in parental cells. Parental A375 and WM115 cells are primary melanoma cell lines whereas WM983B is a metastatic cell line. Hence, acquiring resistance could escalate the activation of signaling mechanisms such as p-IGF1R and p-IR, which could render the A375 and WM115 TDR cells with more metastatic characteristics. It is well established that there are genetic and transcriptomic changes that are acquired while on dabrafenib and trametinib regimen and these changes promote resistance [37,55]. Likewise, our data indicated differential expression of IGF1R and INSR and their high affinity ligands IGF1 and IGF2 (Figure 3A,B).

In this study, we have used BMS-754807, a small-molecule TKI targeting IGF1R and IR in combination with dabrafenib and trametinib on TDR cells. Initially, TKIs only specifically targeting IGF1R were developed. However, owing to high level of structural and functional homology between IGF1R and IR especially in the kinase domain, these tend to target both receptors [56,57]. BMS-754807 combined with dabrafenib and trametinib strikingly reduced the growth of TDR cells (Figure 5A–C). IGF2 is a high affinity ligand for IGF1R and IR compared to IGF1 and insulin [58]. IGF2 ligand can be present in the mature form and unprocessed forms (pro and big) [30]. The unprocessed and mature form of IGF2 displays similar binding potential to IGF1R/IR. IGF2 has the ability to activate IGF1R and IR by autocrine and paracrine mechanisms [59,60]. There are reports showing that BMS-754807 is a more potent inhibitor of the PI3K/Akt over the MAPK pathway [30,61]. Similarly, when TDR cells were exposed to BMS-754807 in the presence of IGF2, we observed an attenuation of p-Akt, but not of p-Erk (Figure 4D). Another likely mechanism for this p-Erk maintenance could be that BMS-754807 stimulates p70S6K1 activity via MEK1/2 and promotes survival as previously reported [62]. We observed a compensatory upregulation in IR and IGF1R when TDR cells were treated with siIGF1R and siINSR, respectively (Figure 6A). Dual knockdown of IGF1R and INSR reduced the growth of TDR cells, providing a rationale for targeting both IGF1R and IR to circumvent the upregulation of the other receptor, which may promote further oncogenesis.

We observed that in WM115 TDR xenograft, BMS-754807 in combination with dabrafenib and trametinib inhibited local tumor growth (Figure 6C). Immunoblotting data from three representative tumors (Figure 6E) correlated with p-IGF1R/IR and p-Akt inhibition as reported in previous data (Figure 4D). Consistent with the literature [63,64], we also observed drug addiction of WM115 TDR tumors to dabrafenib and trametinib, which augmented tumor volume for this group as compared to the vehicle group. Dysregulation of glucose homeostasis would be a major concern for using BMS-754807 in the clinic, as inhibition of the insulin receptor could lead to hyperglycemia. In the short term treatment of BMS-754807, we did not observe signs of toxicity. However, long term tolerance of BMS-754807 in combination with dabrafenib and trametinib remains uncertain. A limitation of our study is that we did not obtain blood glucose levels of mice to assess these effects. In patients with metastatic melanoma, resistant to dabrafenib and trametinib with Type 2 diabetes mellitus, precaution must be taken while dosing and a suitable dose must be established for this patient population. Our results suggest that BMS-754807 along with dabrafenib and trametinib was efficacious in vitro and in vivo. Therefore, further studies investigating the safety and efficacy of combining an IGF1R inhibitor with BRAF and MEK inhibitorsare warranted in cancers harboring BRAF mutations and resistant to targeted therapies.

## 5. Conclusions

In this study, we have characterized trametinib and dabrafenib resistant cells and observed that IGF1R and IR were activated in resistant cell lines. We show differential expression of IGF1R, INSR, along with their ligands IGF1 and IGF2 across the parental and TDR cells. BMS-754807, an IGF1R and IR inhibitor, inhibited proliferation along with p-Akt downstream. Dual genetic knockdown of IGF1R and IR inhibited cell proliferation as compared to siRNA targeting only IGF1R and IR. In vivo examination of combination dabrafenib and trametinib along with BMS-754807 suppressed tumor growth and supported our findings in vitro. This study provides a rationale for targeting IGF1R and IR. Therefore, we propose that combining dabrafenib, trametinib, and an IGF1R/IR inhibitor in patients with BRAF-mutant melanoma who have relapsed or dabrafenib and trametinib therapy regimen could be an effective strategy and warrants further investigation in the clinic.

## Figures and Tables

**Figure 1 cancers-13-05863-f001:**
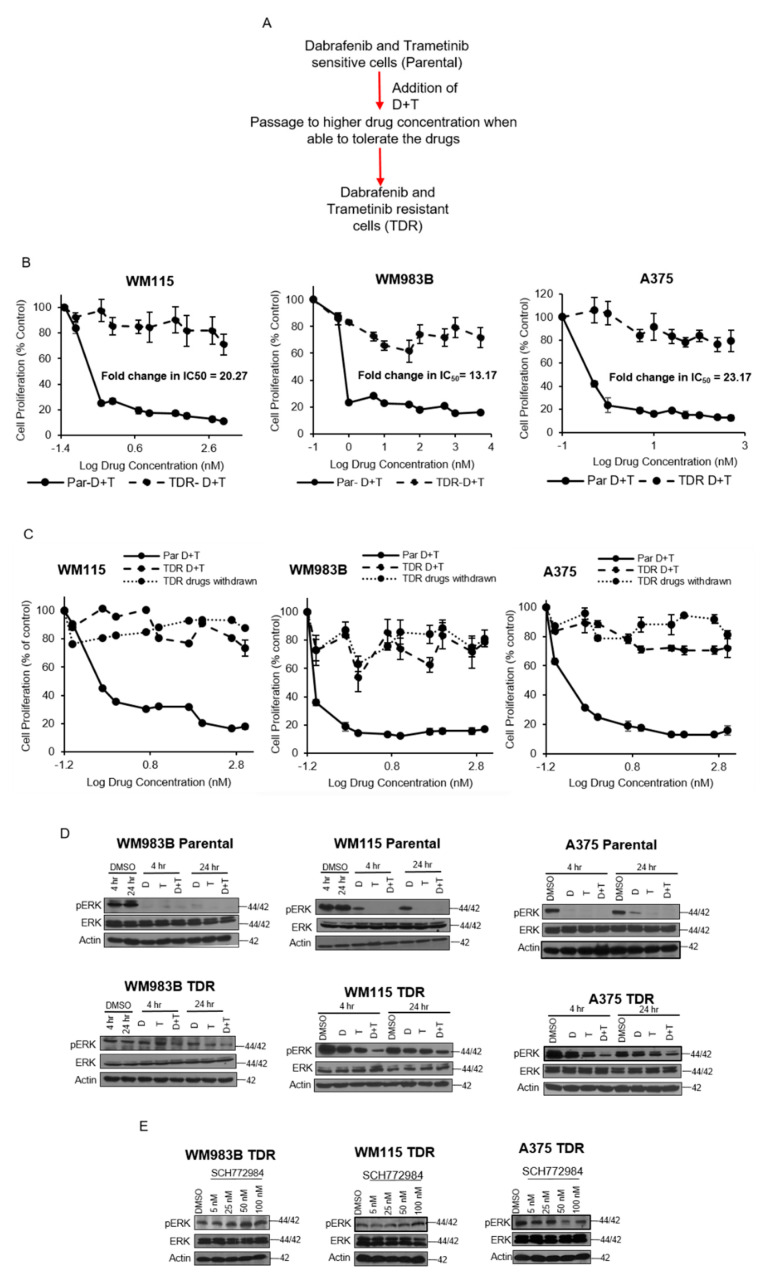
Growth kinetics and characteristics of TDR cell lines. (**A**) Schematic of TDR cell generation. (**B**) WM115, WM983B, and A375 parental and TDR cells (1 × 10^4^ cells/well) were seeded in 96-well plates and treated with increasing concentrations of dabrafenib (D) and trametinib (T) (0.1 nM–5 μM) for 72 h. Cells were treated with MTT for 4 h and absorbance was read at 570 nm. Data reflect three independent experiments and are presented as mean ± SEM. (**C**) Dose response MTT proliferation assay for parental and TDR cell lines maintained in drug-free conditions for 21 days. Cells were plated for 24 h and treated with 72 h with increasing concentrations of dabrafenib (D) and trametinib (T). Cells were treated with MTT. Data reflected as mean ± SEM. (**D**) Immunoblots of WM983B, WM115, and A375 parental and TDR cells treated for 4 and 24 h with dabrafenib (D), trametinib (T), or combination of dabrafenib and trametinib (D + T). Whole cell lysates from the cell lines were analyzed by Western blot using indicated antibodies. Actin served as the loading control. (**E**) Immunoblots for TDR cells treated with increasing concentrations of SCH772984 (5–100 nM) for 24 h. Actin was used as the loading control.

**Figure 2 cancers-13-05863-f002:**
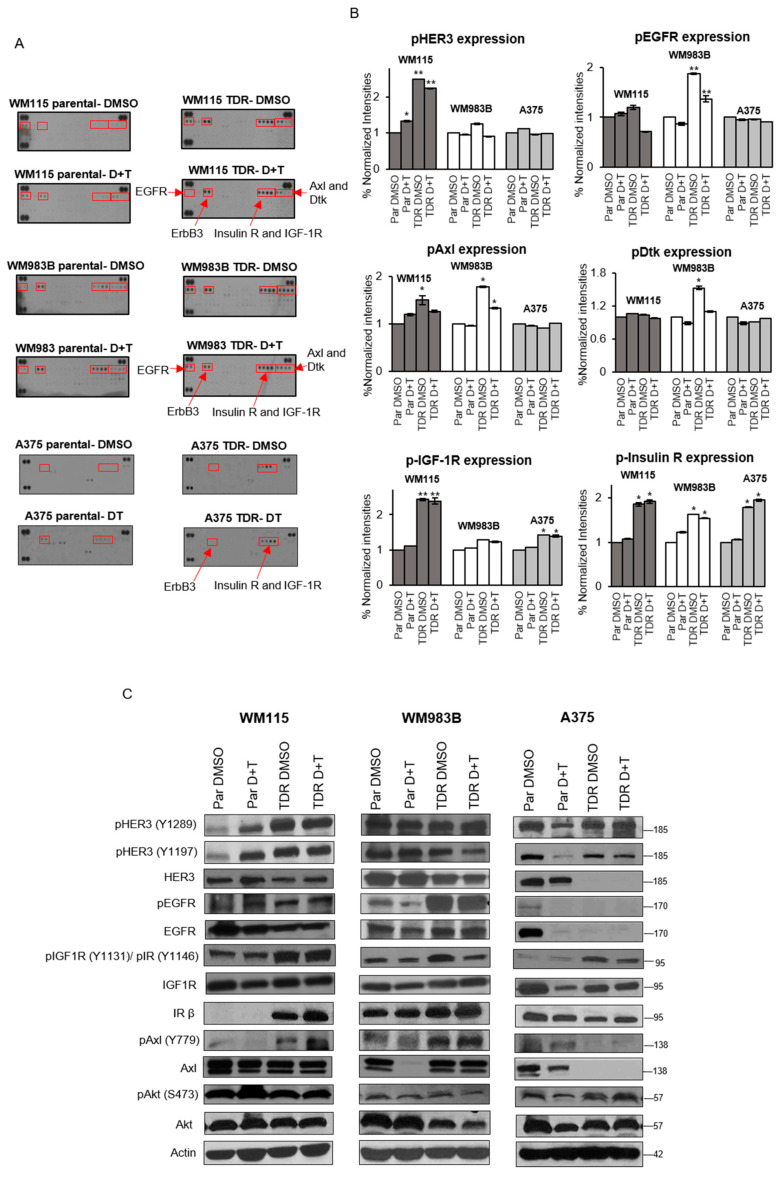
Upregulation of IGF1R and IR kinase activity in BRAF-mutant melanoma cells resistant to BRAF and MEK inhibitors. (**A**) Parental and TDR cells (WM115, WM983B, and A375) either treated with DMSO or combination of dabrafenib (D) and trametinib (T) (WM115: D: 800 nM, T: 200 nM; WM983B: D: 2.4 µM, T: 500 nM; A375: D: 250 nM, T: 12.5 nM) for 24 h. The red boxes indicate the receptors with altered tyrosine kinase activity. Each RTK spotted in duplicate with positive controls in the corner. (**B**) The average signal density of spots normalized to parental DMSO group; *p*-values were computed using GraphPad Prism 7.0. * *p* < 0.05; ** *p* < 0.001 by two-tailed Student’s *t*-test. Data are presented as mean ± SEM. (**C**) Immunoblot analysis of parental and TDR cells treated with DMSO or combination of dabrafenib and trametinib (D + T) for 24 h. Whole cells were analyzed using Western blot with indicated antibodies. Actin served as the loading control.

**Figure 3 cancers-13-05863-f003:**
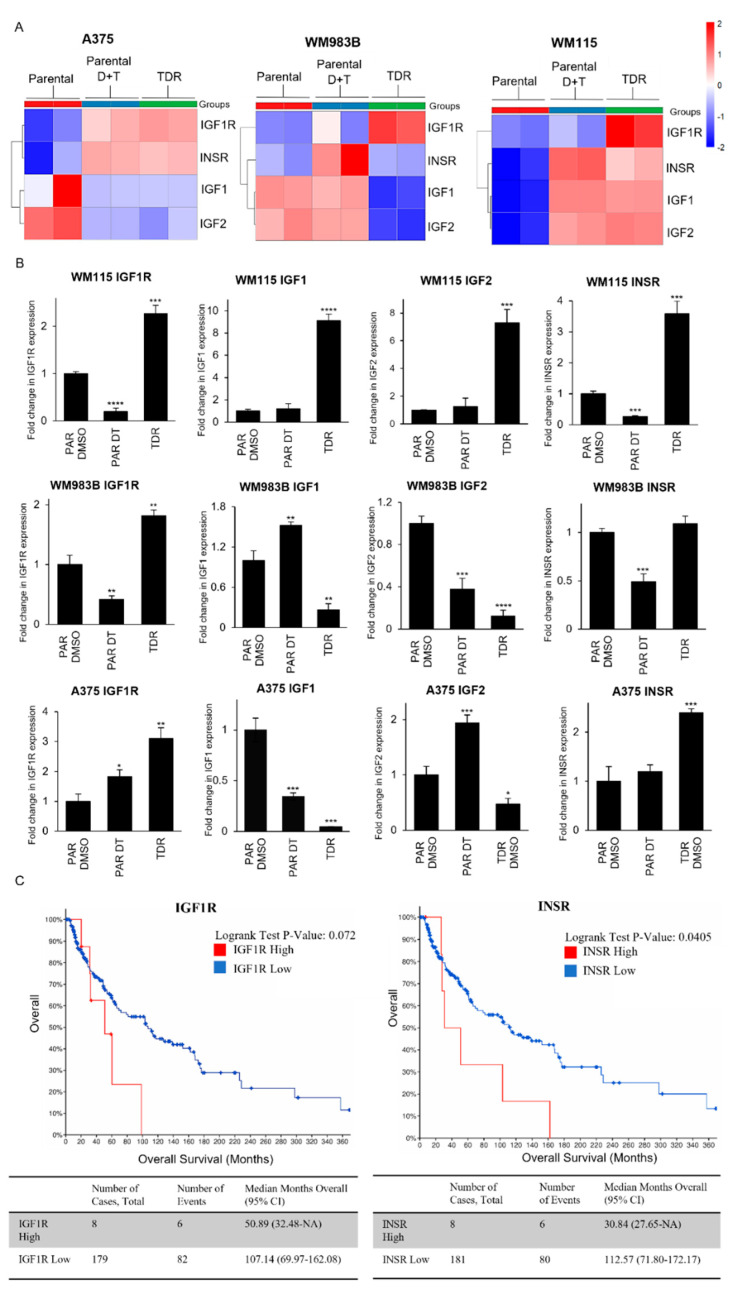
IGF1R and INSR are differentially expressed in TDR cells. (**A**) Heatmap of differentially expressed genes in parental cells, parental cells treated with combination of dabrafenib (D) and trametinib (T) (WM115: D: 800 nM, T: 200 nM; WM983B: D: 2.4 µM, T: 500 nM; A375: D: 250 nM, T: 12.5 nM), or TDR cells. (**B**) Quantification of Actin-normalized mRNA levels using real-time quantitative PCR for IGF1R, IGF1, IGF2, and INSR for parental cells treated with DMSO or dabrafenib + trametinib and TDR following 24 h of treatment. * *p* < 0.05; ** *p* < 0.01; *** *p* < 0.001; **** *p* < 0.0001 by two-tailed Student’s *t*-test. Data are presented as mean ± SEM. (**C**) Kaplan Meier survival curve of skin cutaneous melanoma patients (TCGA, PanCancer Atlas) with data stratified for BRAF mutation along with IGF1R expression *z*-score > 2 (*n* = 8) and IGF1R expression *z*-score < 2 (*n* = 179) (Left panel); and INSR expression *z*-score > 2 (*n* = 8) and INSR expression *z*-score < 2 (*n* = 181) (Right panel).

**Figure 4 cancers-13-05863-f004:**
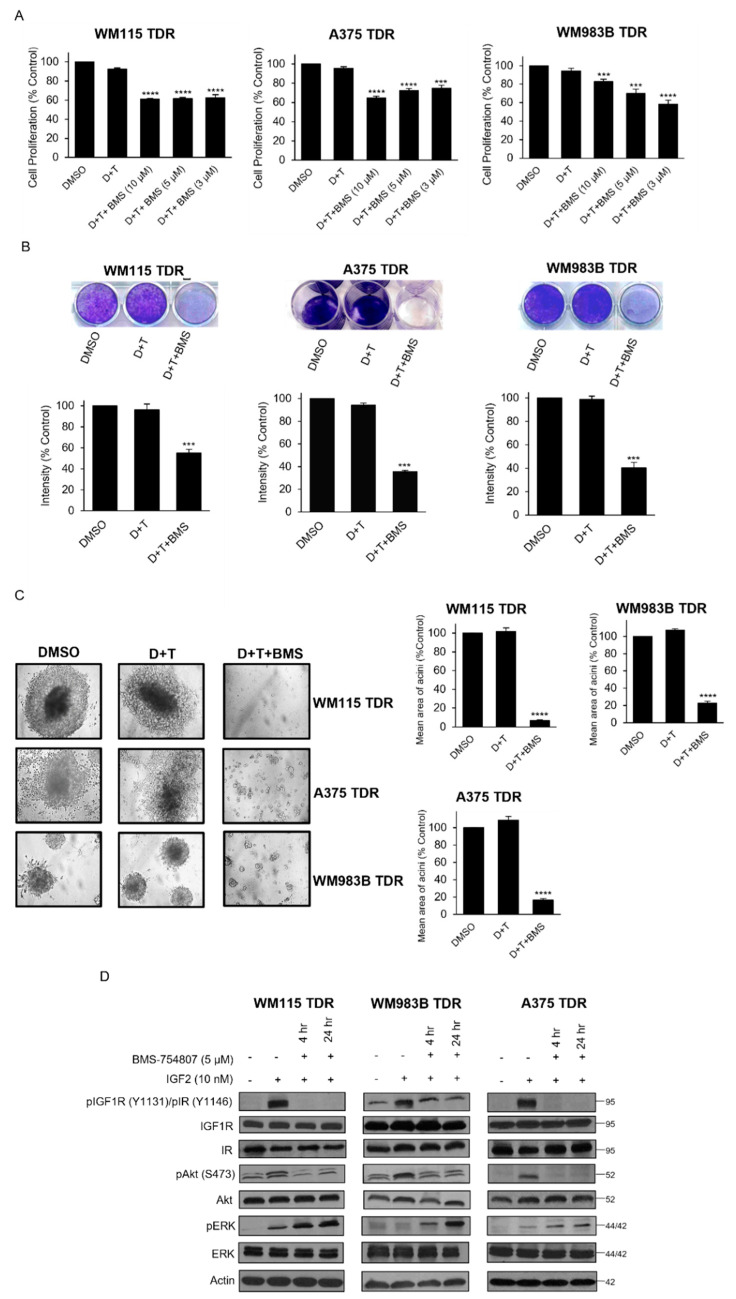
BMS-754807 in combination with dabrafenib and trametinib inhibits cell proliferation and inhibits intracellular Akt (**A**) TDR cells (WM115, WM983B, and A375) plated for 24 h and treated with DMSO, dabrafenib (D), and trametinib (T) (WM115: D: 800 nM, T: 200 nM; WM983B: D: 2.4 µM, T: 500 nM; A375: D: 250 nM, T: 12.5 nM) treated alone or in combination with BMS-754807 (3, 5, and 10 µM) and treated with MTT. The intensities are represented as mean. Error bars: SEM *** *p* < 0.001; **** *p* < 0.0001 by two-tailed Student’s *t*-test compared to DMSO group. (**B**) TDR cells were plated and treated with media containing DMSO, dabrafenib (D) and tramatenib (T) (WM115: D: 800 nM, T: 200 nM; WM983B: D: 2.4 µM, T: 500 nM; A375: D: 250 nM, T: 12.5 nM), or BMS-754807 (5 µM) in combination with dabrafenib and trametinib (D + T + BMS). Media containing drugs were replenished every second day and stained with 0.5% crystal violet on Day 21. Representative images of the well for each treatment (Top panel). Quantification of intensities (Bottom panel) for the wells is represented as mean. Error bars: SEM (*n* = 2 independent experiments performed in triplicates). *** *p* < 0.001 by two-tailed Student’s *t*-test compared to DMSO group. (**C**) TDR cells were seeded on matrigel basement membrane and treated with indicated drugs every second day. Pictures of each well were captured on day 21 (left panel) at 10× magnification. Quantification of mean area of acini (right panel). **** *p* < 0.0001 by two-tailed Student’s *t*-test compared to DMSO group. (**D**) TDR cells were serum starved for 24 h and treated with BMS-754807 (5 µM) along with dabrafenib (D) and trametinib (T) (WM115 TDR: D: 800 nM, T: 200 nM; WM983B TDR: D: 2.4 µM, T: 500 nM; A375 TDR: D: 250 nM, T: 12.5 nM) for 4 or 24 h followed by IGF2 (10 nM) stimulation for 1 h. Whole cell lysates were analyzed by Western blot using indicated antibodies. Actin was used as loading control.

**Figure 5 cancers-13-05863-f005:**
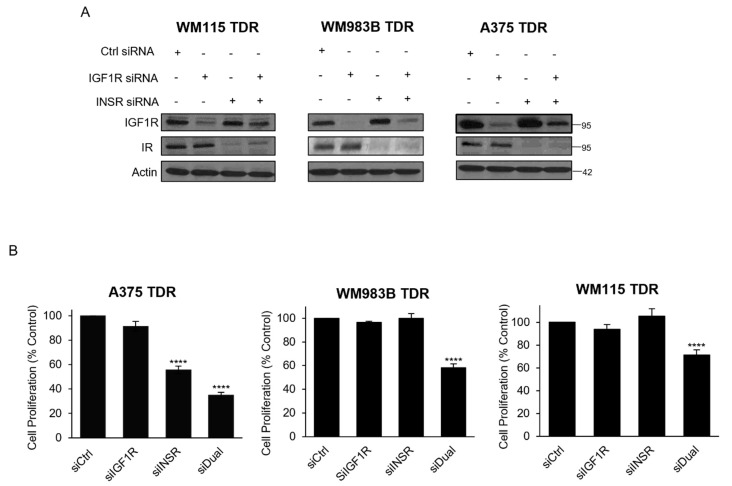
Dual knockdown of IGF1R and INSR reduces proliferation in TDR cells. (**A**) TDR cells were transfected with Ctrl siRNA, IGF1R siRNA, INSR siRNA, or combination of IGF1R and INSR siRNA for 72 h. Cells were collected, lysed, and analyzed for indicated antibodies. Actin was used as loading control. (**B**) TDR cells were transfected as described above, and plated and analyzed for MTT assay. **** *p* < 0.0001 by two-tailed Student’s *t*-test compared to siCtrl.

**Figure 6 cancers-13-05863-f006:**
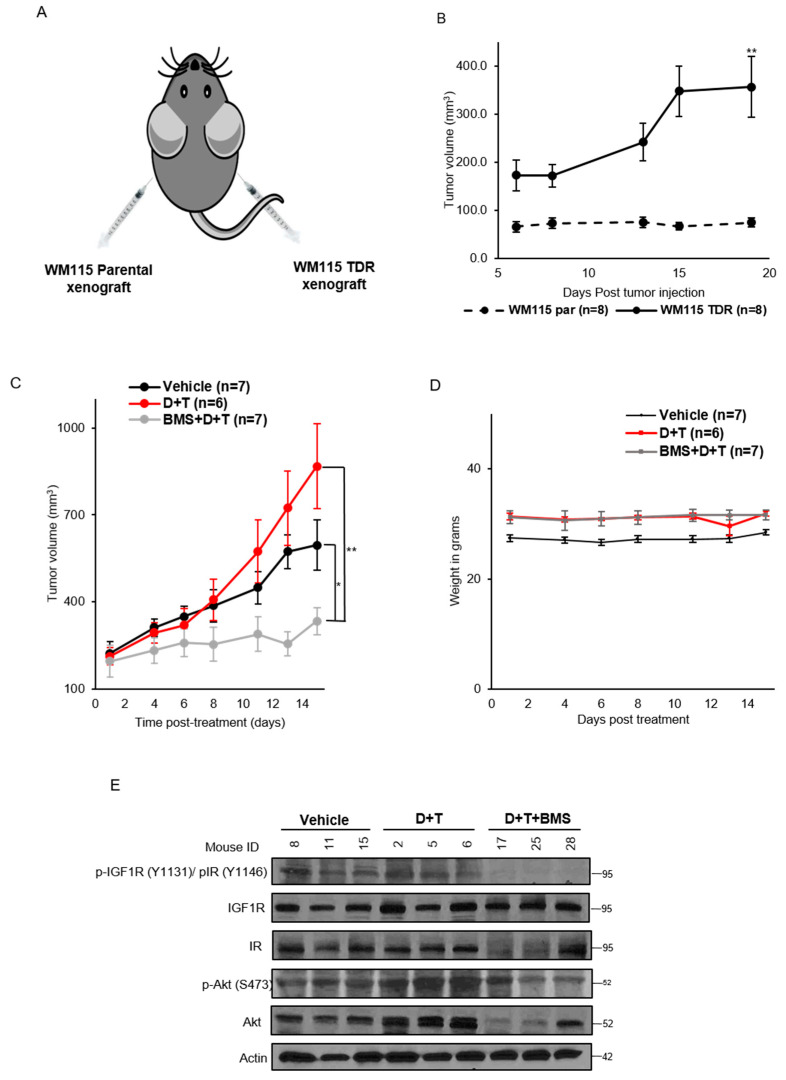
BMS-754807 inhibits tumor growth in WM115 TDR xenograft model. (**A**) Image of WM115 parental and TDR xenograft injection in nude mice. (**B**) Tumor growth kinetics of WM115 parental (*n* = 8) and TDR xenograft (*n* = 8) for 20 days post tumor injection. (**C**) Nude mice were injected with WM115 TDR cells and treated with vehicle, dabrafenib + trametinib (D + T), BMS-754807 (BMS), or the combination (BMS + D + T). Treatment was administered for 2 weeks by oral route. Tumors were measured thrice a week with calipers. Each data point represents the mean tumor volume ± SEM (*n* = 6–7). All data plotted as mean ± SEM. * *p* < 0.05, ** *p* < 0.005 calculated using Student’s *t*-test. (**D**) Mice weight was recorded post treatment with BMS-754807 in combination with dabrafenib and trametinib and represented. (**E**) p-IGF1R/p-IR and p-Akt expressions were assessed in the tumor lysate by Western blot. Actin served as loading control.

**Table 1 cancers-13-05863-t001:** BRAF, PTEN, and CDKN2A status in A375, WM983B, and WM115 BRAF-mutant melanoma cell lines [33].

Cell Line	BRAF Status	PTEN Status	CDKN2A Status
A375	V600E	WT	p.E10, p.E18
WM983B	V600E	WT	p.P63L
WM115	V600D	Loss of heterozygosity [34]	Homozygous Deletion [35]

## Data Availability

The data analyzed in this study was obtained from cBioPortal using Skin Cutaneous Melanoma (TCGA, PanCancer Atlas) dataset. The data presented in this study are available on reasonable request to the corresponding author.

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
