# Peer review of "IGF1R/IR Mediates Resistance to BRAF and MEK Inhibitors in BRAF-Mutant Melanoma"

_cancers, 2021, doi:10.3390/cancers13225863_

Round 1

Reviewer 1 Report

The study reports a novel mechanism of drug resistance in melanomas arboring BRAFV00 mutations. Starting from three BRAF mutated melanoma cell lines (WM115, A375 and WM983) authors generated BRAFi/MEKi resistant clones (TDR) to better explore molecular mechanisms of resistance. Of note, TDR display increased level of phosphorylated IGF1R and IR. and responded in vitro and in vivo to combination of BRAFi/MEKi together with IGF1Ri/IRi This finding is of potential clinal relevance and identify novel co-targets of BRAFV600 mutations

Major Point:

  • Paragraph 2.10 in vivo xenograft study: the author should better define timing and doses of xeno tratment
  • Figure 1: pERK western blot for WM983B TDR and A375 TDR cell lines of figure 1D and for all cell lines in figure 1E are suboptimal. The whole blots relative to figure 1 that are reported in figure S5 and S6 display a significant amount of non-specific signal. Moreover, I suggest ordering the blots in figure S5 as in figure 1D.
  • Figure 2: In the graphs of figure 2B relative to p-IGFR-1R and p-Insulin R expression, the bars of ‘A375 Par D+T’ is lower than ‘A375 Par DMSO’ at variance with Figure 2A. Furthermore, signal density should be normalized to parental DMSO group and not to parental DMSO-treated group as reported in Figure Legend. Please check the legend. For quality purpose I would suggest to improve pHER3 (Y1289), pEGFR, pIGF1R and pAxl western blots for each condition of all cell lines; HER3 for each condition of WM115; pIR1β for each condition of WM115 and WM983B. The whole blots relative to Figure 2C reported in figure S7 have significant nonspecific signal.
  • Figure 3: The RNA-seq analysis reported in Figure 3A is not clear. Results relative to ‘parental D+T’ cells are not commented in the text, for example: Why are IGF1R and INSR upregulated in A375 Parental D+T cells? Why is INSR upregulated in WM983B Parental D+T cells and not in WM983B TDR cells? Furthermore, concentration and time of Dabrafenib and Trametinib treatment are not reported. In Figure 3B results relative to ‘parental D+T’ cells are again not well commented in the text and the concentration of Dabrafenib and Trametinib treatment is not reported. Moreover, RNA-seq result and Real-Time analysis are discordant forWM115 parental D+T, WM983B parental D+T INSR, A375 parental D+T IGF2.
  • Figure 4A: the concentration of Dabrafenib and Trametinib is missing. I suggest checking the asterisks in the graph and using a unique code for p value in all figures of the article: *p < 0.05, **p < 0.01, ***p < 0.001, ****p < 0.0001. Moreover, it is not clear to me the reasons why WM983B cells treated with BMS 3µM are less proliferative than WM983B cells treated with BMS 10µ
  • Figure 4B: Again, the concentration of dabrafenib and trametinib is not reported and I suggest checking the asterisks in the graph and using a unique code for p value in all figures of the article: *p < 0.05, **p < 0.01, ***p < 0.001, ****p < 0.0001.
  • Paragraph 3.6: I suggest implementing the in vivo analysis with an additional cell line (Wm983B orA375).
  • Line 423 – 424: Upregulation is confirmed only by RTK array analysis.
  • Line 454 – 456: blot of phosphorylated protein should be improved.

Minor Point:

  • Line 11: “but is” instead of “butis”, please correct the text.
  • Line 23: I could suggest MEK and BRAF inhibitors respectively, instead of BRAF and MEK inhibitor.
  • Line 44: “and” is not necessary in the sentence, please correct the text.
  • Paragraph 2.1 reagents: Linsitinib, BMS-536924 and GSK1838705A are not reported in Materials and Methods section.
  • The catalogue number is missing for: IGF2 (line 90); RTK Array Kit (line 112); growth factor-reduced matrigel (line 138); Pierce BCA Protein Assay Kit (line 148); anti-rabbit IgG-HRP (line 152); RNA extraction mini kit (line 165 and line 184); iScript cDNA Synthesis kit (line 168); NEBNext® Poly(A) mRNA Magnetic Isolation Module (line 186); NEBNext® Ultra II Directional RNA Library Prep Kit (line 189); Matrigel (line 201);
  • Line 117: “WM115 TDR” instead of “WM115TDR”, please correct the text.
  • Line 119: is not reported how many and which concentration are used for the experiment but only the minimum concentration (0.1 Nm) and maximum concentration (10 µM).
  • Line 201: It has been used also left flank with WM115 parental cells. Please add this part.
  • Line 205 – 206: I suppose that PEG400: water 1:1 is used also for ii condition but is not reported. Please check the text.
  • Line 273: “with” instead of “with with”, please correct the text.
  • Line 272 – 274: WM115 parental cells exhibited inhibition of p-IGF-1R and not WM983, please check.
  • Figure S2: Some results of the Real-Time PCR analysis are missing. Please complete the figure with missing data; in alternative organized differently the figure panel. Moreover, there is an error in p value asterisks: ***p < 0.001 instead of ***p < 0.01, please correct the text.
  • Line 337: “WM115” instead of “Scheme 115”, please correct the text.
  • Figure 5B: statistical analysis is missing in figure legend, please check.
  • Figure 6C: the treatment with only BMS-754807 is not reported in graph.
  • Line 382: 2 or 3 tumours?
  • Line 393: calculated instead of calculated calculated, please correct the text.

Reference:

  • Line 44 – 47: The percentage of frequency of BRAFV600E mutation is incorrect; the reference number 7 reported that 84.6% of melanomas have BRAFV600E mutation instead of 50-70%. Please check the reference.
  • Line 193: the reference number 32 does not report to iGEAK. Whereas I suppose that reference number 64, which is not reported in the text, should be insert as iGEAK reference, please check.
  • Line 225: The reference number 33 is not reported, please check.

Supplementary:

  • Figure S3: some data are missing, please perform the experiment.
  • Figure S8: I suggest performing again pIGFR1R/pIR for WM115 and A375; pERK for WM983B and A375. There is too much non-specific signal.
  • Figure S10: I suggest performing again pIGFR1R/pIR and pAkt for all cell lines. There is too much non-specific signal.

Reviewer 2 Report

Having read the manuscript "IGF1R/IR mediates resistance to BRAF and MEK inhibitors in BRAF-mutant melanoma" I have the following comments:

  1. It is not clear from the Materials and methods section as to how you performed the Western blots or qualified the expression of protein in such. The supplementary materials show some oversaturated blots which densitometry is unable to quantify with any accuracy.  Can you please elaborate on this.  Also your ERK Ab shows multiple bands which differs to most other commercial ERK Abs.
  2. Why did you not examine a BRAFWT melanoma cell line to observe if IR or ILGFR play a role in rescuing these cells from MEK inhibition as well?
  3. In the methods why is  the Greek symbol µ bolded?
  4. In general the Materials and methods section is very brief and more information on some experimental procedures are needed, eg. what is the vol of cell suspension used in the in vivo study?
  5. Did you reprove your blots?  If so please list how this was done, and how were protein the levels quantified?  It is not clear if this was done or not.
  6. In the growth kinetic studies for how long was MTT added to the cells before the absorbance was recorded?
  7. Please mention in full the name that is abbreviated in the manuscript before or after the abbreviation.
  8. How long were the cultures exposed to crystal violet?
  9. You should write all Latin words in italics eg. in vitro, in vivo etc
  10. References 12, 13, 15, 21, 32, 33, 34, 46, 54, 57 60, 61 & 64 are not correctly written, please correct.
  11. There are many instances of poor grammar and spelling in this manuscript which require editing by a native English speaker.

Reviewer 3 Report

The authors generated TDR cell lines in 3 different BRAF-mutant melanoma lines, and then used a RTK array strategy to identify potential pathways related to BRAFi/Meki resistance.  Among the heterogeneous RTK responses, the IGR1R/IR pathway was identified as a potential shared mechanism.  The authors further validated these findings and showed potentials to use BMS-754807 together with Trametinib and Dabrafenib to enhance treatment efficacy in a xenografted mice model. The heterogenous data reveal complex biological responses in the MAPK kinase resistance pathways which sometimes are difficult to summarize.  The authors presented sufficient amount of convincing data to support their conclusion.  

A few minor points to address:

1) Figure 3. survival curve, (also line 304-305), state that only BRAF-mutant patients are included in the analysis.  Also does this only related to BRAF-mutant patients? what happened to BRAF-wt patients? Please include statistical analysis method in the "method" section.  

2) Figure 4, what happens if the cells are treated with only BMS? This control should be added.

3) IGF1R variants are found in patients with multiple melanomas (Yuan et al., 2020, IJMS 21(5):1776) which should be cited. 

Round 2

Reviewer 1 Report

The manuscript is significantly improved.. The author response to my query are completely satisfactory.